# Goal Orientations of Secondary School Students and Their Intention to Practise Physical Activity in Their Leisure Time: Mediation of Physical Education Importance and Satisfaction

**DOI:** 10.3390/healthcare11040568

**Published:** 2023-02-14

**Authors:** Francisco Javier Pérez-Quero, Antonio Granero-Gallegos, Antonio Baena-Extremera, Raúl Baños

**Affiliations:** 1Department of Education, University of Almeria, 04120 Almeria, Spain; 2Health Research Centre, University of Almeria, 04120 Almeria, Spain; 3Department of Musical, Plastic and Corporal Expression, Faculty of Education Sciences, University of Granada, 18071 Granada, Spain; 4Department of Musical, Plastic and Corporal Expression, Faculty of Social and Human Sciences, University of Zaragoza, Campus de Teruel, 44003 Zaragoza, Spain; 5Faculty of Sport, Autonomous University of Baja California, Tijuana 22390, Mexico

**Keywords:** motivation, satisfaction, task orientation, ego orientation, physical education

## Abstract

The aim of this study was to analyse the mediating role of Physical Education importance and satisfaction/fun between the dispositional goal orientations of secondary school students and their intention to partake in leisure time physical activity. The research design was descriptive, cross-sectional, and non-randomized. In total, 2102 secondary school students participated (*M*_age_ = 14.87; *SD* = 1.39) (1024 males; 1078 females). The scales used were the Perception of Success Questionnaire, Importance of Physical Education, Satisfaction with Physical Education, and Intention to Participate in Leisure Time Physical Activity. Structural equation models with the latent variables were also calculated. The results highlight that Physical Education satisfaction/fun has a mediating effect between task orientation and the intention to practice physical activity during leisure time.

## 1. Introduction

Physical Education (PE) in Spain has significant importance in Organic Law 3/2020, of 29 December [1]. This law regulates non-university educational teaching in the different age groups and encourages the education system and PE to help people incorporate sports practice into their lives and thus contribute to an active and healthy lifestyle. Moreover, this law adds an additional provision that was not contemplated in the previous one, Organic Law 2/2006, of 3 May [2]. This provision refers to the promotion of physical activity and healthy eating so that they are part of the behaviour of children and young people. This line mentions that the administrations will promote the daily practice of sports and physical exercise by students during the school day, under the terms and conditions that, following the recommendations of the government agencies, guarantee adequate development to promote a healthy and autonomous life, to promote healthy eating habits and active mobility, reducing sedentary lifestyles [1]. Despite this, adolescence is a stage in which the practice of physical activity is still largely neglected; in addition, there is a high rate of physical inactivity in this population [3,4], which is why it is an important issue to address in the scientific field. 

The sedentary lifestyle is a global problem, although in Spain, it is particularly associated with various psychological [5] and physical pathologies [6]. In this regard, it should be noted that Spain is the European country with the highest prevalence of mental health issues, with 20.8% of Spanish adolescents suffering from mental disorders such as depression, anxiety, and behavioural disorders, among others, according to the United Nations Children’s Fund [7]. In addition, there is a prevalence of being overweight or obese amongst more than 2.5 million Spanish children and adolescents [8]. According to the National Institute of Statistics [9], the data on physical inactivity are really worrying—31.8% of secondary and upper secondary school students claim to be inactive during their leisure time. The problem is exacerbated by adolescents claiming that they have no intention of practising any physical sports activity in their leisure time, apart from school lessons [10]. Consequently, work needs be done on increasing the students’ intention to do physical exercise outside school, and since research on secondary school students has revealed that satisfaction and enjoyment of PE, and the importance given to this subject, can be predictive variables of their intention to practise physical exercise apart from school lessons [11,12,13,14], it is advisable to address the mediating role of these variables (satisfaction/fun, importance) between other variables such as the dispositional goal orientations (i.e., task orientation, ego orientation) with which the students approach this subject (i.e., PE) and their intention to practise physical exercise apart from school hours.

The Theory of Planned Behaviour [15] considers “intention” to be the most predictive factor of future behaviour disposing a person towards practising physical activity [16]. According to Ajzen [15], intention can be influenced by three factors: one that reflects the social influence, the subjective norm, understood as the pressure adolescents perceive from their peers or teacher, and two other factors of a personal nature—the attitude towards the behaviour, in which the performance of a certain behaviour is evaluated either positively or negatively, and the perceived behavioural control, which directly predicts the behaviour depending on whether it is under voluntary control or not, and whether there are differences between the control that the person believes they have and that which they actually have. However, it should be noted that there is a gap between intention and behaviour, that is, not every intention to perform a behaviour becomes a behaviour itself [17]. Regarding the intention to be physically active, studies have shown this gap exists between the intention to be physically active and developing a positive behaviour towards physical activity [17,18]. However, the intention is the proximal antecedent of the enactment of a behaviour [19]. In this line, the intention of adolescents to practice physical activity outside school hours has been related both to the importance and usefulness they give to PE [20] and to the satisfaction they experience with the subject [10]. For this reason, we consider it important to investigate how we can act regarding PE, in order to increase the intention of adolescents to be physically active in their free time.

Adolescence is a critical stage for students to develop positive attitudes towards physical activity and sport, with PE playing a fundamental role in generating active lifestyles [12,21]. If students consider PE important and find it useful for their routine life, they are more likely to generate active behaviours in the future, increasing their intention to be physically active in their free time, apart from PE lessons [13,20,22]. In turn, it is important that adolescents have fun and feel satisfied in PE lessons, since satisfaction predicts how they perceive both the importance and usefulness of PE and the intention to be physically active in their leisure time, apart from PE lessons [3,12]. Conversely, if they experience dissatisfaction or boredom, it will detract from the subject and so decrease active behaviours outside of school [23].

The Subjective Well-Being Theory is a theoretical construct that analyses the perception of human beings regarding their satisfaction with life generally, as well as in specific areas of life such as family, friends, school, sports practice, etc. [24]. PE can be considered a specific subject for adolescents in which they might make judgments in terms of satisfaction/fun or dissatisfaction/boredom based on how they learn and experience the subject [10]. In this vein, Balaguer et al. [25] designed an instrument that measured satisfaction with the school, which was later adapted to PE by Baena-Extremera et al. [26], in order to measure the satisfaction and enjoyment experienced by students with PE, in this specific area of adolescent life. This enjoyment with PE acts as an excellent predictor of both the intention to practice physical activity outside PE lessons [9] and their participation in physical sports activities [27]. Thus, generating enjoyable learning environments in PE can be an important variable in approaching the goal of adolescents continuing to practise sports outside of school hours [26]. How adolescents approach their goal perspectives can have a great impact on how satisfied they are with PE [26] and on their intention to be physically active [28].

The Social Cognitive Theory within Achievement Goal Theory describes the two perspectives that predominate in PE—one being task-oriented and the other ego-oriented [29]. According to the author, when an individual judges their own level of ability, an orientation towards the task prevails; in contrast, when the individual compares their ability level to others, an ego orientation predominates. Various studies have linked task orientation with greater fun and enjoyment of PE, whereas students who have an ego orientation suffer more from boredom and do not enjoy practising the subject [26,29,30]. Dispositional goal orientations in PE are so relevant that when adolescent students find themselves in a task-oriented climate, they tend to perceive PE classes as more enjoyable and are more active outside of school [31].

As described above, PE plays an important role in a student’s intention to be physically active outside of school hours because it involves self-determined motivation and the importance and usefulness that adolescents find in the subject [13,20,22,32], the enjoyment and fun they experience [10], and the type of dispositional goal orientation that exists in the class [28]. However, to the best of our knowledge, there are no studies of a predictive model that includes all these variables. Therefore, this research aims to analyse the mediating role of satisfaction with PE and the perceived importance and usefulness of PE between the dispositional goal orientations of secondary school students and their intention to practise physical activity in their leisure time. After reviewing the scientific literature, we have considered a hypothesized model (see Figure 1) containing the following hypotheses: (H1) the importance and usefulness that students perceive in PE will have a negative mediating effect between ego orientation and the intention to be active outside of the school environment; (H2) the importance and usefulness that students perceive in PE will have a positive mediating effect between task orientation and the intention to be active outside of the school environment; (H3) satisfaction with PE will have a negative mediating effect between ego orientation and the intention to be active outside of the school environment; (H4) satisfaction with PE will have a positive mediating effect between task orientation and the intention to be active outside of the school environment.

## 2. Materials and Methods

### 2.1. Design and Participants

The study design was cross-sectional. An a priori analysis of the statistical power of the sample size was carried out [28]; it was estimated that a minimum of 1970 students were necessary for effect sizes of f^2^ = 0.126 with a statistical power of 0.99, and a significance level of α = 0.05 in a structural equation model with five latent variables and 23 observable variables [33]. The study involved a total of 2102 secondary school students (1024 male) from 18 secondary schools in Andalusia. Distribution by course was as follows: 34.9% studied in the second grade of Compulsory Secondary Education (CSE); 16.9% in third grade of CSE; 23.6% were fourth graders of CSE; and 24.6% were in their first year of high school (bachillerato). The age ranged from 12 to 19 years (*M* = 14.87; *SD* = 1.39). These students had a medium economic socio-level, with a 3% rate of dropout, and 7% of pupils in the classes were foreign. The classes were mixed (boys and girls), and all pupils had PE as a compulsory subject (two sessions of 60 min/per week). In addition, 28.3% of the students participated in sporting competitions outside of school during the week.

### 2.2. Procedure

After obtaining authorization from the schools to carry out the research, the students were informed of the study objective and their rights as participants in it, how to answer the questionnaire, that the answers would be kept anonymous, that they would not affect their grades in the subject, and that they could stop participating in the study at any time. The data were collected in person by a researcher during the PE class, after previous agreement with the teacher. The students had 15 min to answer the questionnaire, and the PE teacher was not present in the classroom. Prior consent was obtained from the parents/legal guardians of all the participants included in the study. Approval for the research protocol was obtained from the University Ethics Committee (Ref: 19002018) and was carried out in accordance with the Declaration of Helsinki.

### 2.3. Instruments

Perception of Success Questionnaire (POSQ). The Spanish version [34] adapted to PE [35] of the original POSQ [36] was used. The scale consists of 12 items that measure the students’ dispositional goal orientations in PE classes: task orientation (six items) and ego orientation (six items). Responses are collected on a five-point Likert scale ranging from 1 (strongly disagree) to 5 (strongly agree). A higher score indicates higher dispositional goal orientation (i.e., task orientation, ego orientation). In the present study, the CFA model presented the following goodness-of-fit indices: χ^2^/df = 3.45, *p* < 0.001; CFI = 0.97; TLI = 0.97; RMSEA = 0.059 (90%CI = 0.052, 0.066), SRMR = 0.038. The reliability obtained was: task orientation (McDonald’s Omega, ω) = 0.88; ego orientation, ω = 0.91.

Importance of Physical Education (IPE). The version by Moreno et al. [21] was used, which evaluates the importance and usefulness students give to PE through three items. Responses are collected on a four-point Likert scale ranging from 1 (strongly disagree) to 4 (strongly agree). A higher score indicates higher importance of PE. In the present study, the CFA model presented the following goodness-of-fit indices: χ^2^/df = 2.45, *p* = 0.126; CFI = 0.98; TLI = 0.96; RMSEA = 0.045 (90%CI = 0.038, 0.051), SRMR = 0.042. The reliability obtained was: ω = 0.78.

Satisfaction with Physical Education (SSI-PE). The satisfaction/fun subscale from the Spanish version of the SSI-PE [26], part of the original Sport Satisfaction Instrument [25,37,38], was used. It consists of five items that measure satisfaction/fun with PE classes. Responses are collected on a five-point Likert scale ranging from 1 (strongly disagree) to 5 (strongly agree). A higher score indicates higher satisfaction with PE. In the present study, the CFA model (satisfaction/fun) presented the following goodness-of-fit indices: χ^2^/df = 1.01, *p* = 0.400; CFI = 0.99; TLI = 0.99; RMSEA = 0.013 (90%CI = 0.002, 0.042), SRMR = 0.004. The reliability obtained was: ω = 0.94.

Intention to partake in leisure time physical activity (Intention-PLTPA). The Spanish version [39], based on Chatzisarantis et al. [36], Ajzen and Madden [16], and Ajzen and Fishbein [40], was used. This one-dimensional scale, consisting of three items, measures the intention of secondary school students to be physically active in their leisure time (outside of school). Responses are collected on a 7-point Likert scale ranging from 1 (very unlikely) to 7 (very likely). A higher score indicates higher intention to partake in leisure time physical activity. In the present study, the CFA model presented the following goodness-of-fit indices: χ^2^/df = 1.78, *p* = 0.171; CFI = 0.99; TLI = 0.99; RMSEA = 0.003 (90%CI = 0.001, 0.039), SRMR = 0.003. The reliability obtained was: ω = 0.93.

### 2.4. Data Analysis

Descriptive statistics and correlations between the analysed variables were estimated with the SPSS v.28 programme. In addition, the McDonald’s omega coefficient was calculated for each of the variables, considering values >0.70 as being indicative of good reliability [41]. The SEM was controlled for gender, age, and sports competition outside of school during the week a two-step structural equation model (SEM) was carried out with AMOS v.26 [42] to analyse the predictive relationships between the students’ dispositional goal orientation and the intention to perform physical sports practice in their leisure time, analysing the mediating role of importance and satisfaction with Physical Education. In the first step, referred to as the measurement model, the robustness of the bidirectional relationships between the variables that make up the model was analysed. In the second step, the predictive effects between the variables were determined. In the event of the multivariate normality assumption being violated (Mardia’s coefficient = 217.23; *p* < 0.001), the analysis was performed using the maximum likelihood method and the 5000-iteration bootstrapping procedure [42]. To evaluate the model’s goodness of fit, chi-square and degrees of freedom (χ^2^/df) values <5.0 were considered acceptable, as were CFI (Comparative Fit Index) and TLI (Tucker–Lewis Index) values >0.90, in conjunction with values up to 0.80 for the SRMR (Standardized Root Mean Square Residual) and RMSEA (Root Mean Square Error of Approximation) [43,44]. To analyse the direct and indirect effects, the proposal of Shrout and Bolger [45] was followed; thus, the indirect effects (i.e., mediated) and their 95% CI were estimated with the bootstrapping technique, and the significant indirect effect (*p* < 0.05) was considered if its 95% CI did not include the zero value.

## 3. Results

### 3.1. Preliminary Results

The descriptive statistics and the correlations between the latent study variables are presented in Table 1. First, it is notable that the mean values for the dispositional goal orientation are higher in the task orientation than in the ego orientation. The rest of the variables present moderately high values, considering the measurement range. Regarding the correlations, all are statistically significant, and the task orientation presents closer relationships with the ego orientation and satisfaction with PE. The correlations between the ego orientation and the rest of the variables are low, while the close relationship between satisfaction with PE and the intention to PLTPA is striking, as is the relationship between the importance of PE and satisfaction with PE. Regarding the reliability of the scales, all presented values are higher than 0.70.

### 3.2. Main Results

In step 1, the model presented acceptable goodness-of-fit values: χ^2^/df = 3.024, *p* < 0.001; CFI = 0.97; TLI = 0.97; RMSEA = 0.040 (90%CI = 0.037; 0.043), SRMR = 0.056. In step 2, the predictive SEM model also presented an acceptable fit: χ^2^/df = 4.580, *p* < 0.001; CFI = 0.95; TLI = 0.94; RMSEA = 0.053 (90%CI = 0.050; 0.056), SRMR = 0.069. The SEM achieved an explained variance of 37% for intention-PLTPA, 20% for satisfaction with PE, and 9% for importance of PE. After controlling for gender, age, and sports competition outside school lessons during the week, in the SEM (Figure 2), it was observed that ego orientation has no effect on the intention to practice physical activity in one’s leisure time or outside the PE classes, since it does not present statistically significant relationships with any of the variables (i.e., importance of PE, satisfaction with PE, intention-PLTPA). In contrast, the direct relationships were significant between task orientation and the importance of PE, satisfaction with PE, and intention-PLTPA, as were the direct effects of the importance of PE and satisfaction with PE on intention-PLTPA. It is worth noting the mediating effect of satisfaction with PE between task orientation and intention-PLTPA (0.19), as it increases the effect between the latter two variables. The mediating effect of the importance of PE between task orientation and intention-PLTPA is less relevant (0.05) than that of satisfaction with PE. In addition, the overall effects of task orientation on intention-PLTPA were higher with the mediating effect of satisfaction with PE (see Table 2).

## 4. Discussion

The objective of the present research was to analyse the mediation of the importance and usefulness of PE and satisfaction with PE between dispositional goal orientations and the intention of secondary school students to practise physical activity in their leisure time. The main results show that task orientation has a direct and positive effect on the intention to practice physical activity in one’s leisure time. The total effects on this variable are increased, above all, with the mediation of the satisfaction with PE that the student experiences. Ego orientation is not significantly related to any of the variables studied.

Few studies have analysed the prediction of goal orientations on the intention of adolescents to practice physical activity in their leisure time. The present work shows that task orientation directly and positively predicts the intention to practice physical activity during their leisure time in the future, while no significant results were obtained in those students with an ego orientation. Similar results were obtained by Franco et al. [28]—in their study, only task orientation predicted the intention to practice physical activity in the future. Several studies have highlighted how important it is for PE teachers to design their classes to encourage task-oriented learning climates—this can be decisive, both for achieving positive results in PE classes and for increasing active lifestyles outside of school [20,46]. These results could be due to the fact that when students have a task-oriented disposition to achieve their goals, they believe the success of the final result depends on their own effort, interest, and self-learning [47,48,49], with these behaviours being related to a lower probability of them abandoning sports practice in their leisure time [31].

In addition, the results of the present study show that the total effects of dispositional goal orientation on the intention to practice physical activity in one’s leisure time are increased with the mediation of adolescents finding PE important and useful but, above all, of them enjoying the subject. We are not aware of any research that has analysed the mediating effect of these variables. Focusing on the mediating role of satisfaction with PE, several studies have found that when a student has fun and feels satisfied with PE classes, their intention to be active in their leisure time will increase [10,20,28], with task orientation being a strong predictor of satisfaction with PE [12]. This relationship could be due to the fact that when adolescents are task-oriented, they tend to make an effort in their personal development, focusing their satisfaction on self-improvement [47,50,51], thus increasing their satisfaction with PE [12,48]. Various authors have highlighted the importance of strengthening student satisfaction and motivation in order to continue physical activity outside the school classroom [3,32,52,53,54]. In this way, students will be enriched by positive experiences in the PE classroom, a fundamental aspect for their commitment to healthy and active lifestyle habits [55,56]. Conversely, the accumulation of negative physical activity experiences can lead to habitual physical inactivity [3,53,54,55,57].

In the case of the importance and usefulness of PE for students, we are likewise unaware of any research that has analysed the mediating effect of these variables. In this regard, we can say that our results agree with those obtained by other studies, which related this variable with the intention to be physically active [20,21,22,23] and with task orientation [58,59]. This could be due to the fact that when students are task oriented, they use internal sources to judge their motor ability, and when they fail to achieve the results, they respond with increased effort and greater persistence to perform the activity until they are successful [60]. In this way, they will give greater importance to PE, finding it more useful for solving everyday situations [61].

Finally, a series of limitations and strengths should be mentioned. In terms of limitations, mention that the present study only measured the intention, not the actual behaviours, that is, the intentions of adolescents to be physically active do not always translate into active behaviours. It is worth noting the cross-sectional research design, since when it is carried out in a determined time, it is not possible to analyse the behaviour of the subjects during a determined time. Another limitation of the study was that the measurements and the results obtained were carried out through questionnaires, this could condition the responses of the participants due to the search for social desirability, causing a bias in the research. Finally, it is worth mentioning that the sample was not representative, so the results cannot be generalized to Spanish adolescents. However, regarding the study’s strengths, the size of the sample and the type of statistical analysis can be highlighted, since the SEM has been carried out with latent variables. As future research, we suggest that longitudinal or experimental studies should be carried out, in which it is analysed over time if the intention to be physically active becomes behaviour, and to analyse which are the most influential variables in this transformation. Finally, the research topic addressed in this study is of great interest, namely, proposing a possible solution to physical inactivity approached through the subject of PE.

### Practical Implications

The results of the present study underline the importance of task orientation among adolescent students and, in turn, that they feel satisfied with PE and that they understand its importance and usefulness, thus encouraging the intention to continue with physical activity in their leisure time. For this reason, it is recommended that secondary school PE teachers highlight the importance of PE in their classes and the usefulness that students can get from it in their day-to-day lives—for example, by creating an educational blog about PE [62], encouraging walking or cycling to school [63], or creating fun classes and new content [64,65], among others. Furthermore, it is recommended that they design their classes to support student autonomy, with sessions that are motivating and that focus on self-improvement, since this can help increase enjoyment and satisfaction with PE, and thus the intention of leading active lifestyles [66,67,68,69,70].

## 5. Conclusions

In summary, it can be stated that a task-oriented disposition in students increases their intention to practise physical activity in their leisure time. Furthermore, this intention to lead an active lifestyle increases especially when students have fun and are satisfied with PE, although it is also necessary that students consider the subject important and useful. Finally, secondary school PE teachers should incorporate methodological strategies in their classes that can be extrapolated to their adolescent students practising physical sports activity in their leisure time, promoting the use of the surrounding environment (whether natural or urban), as well as encouraging student autonomy in deciding what activities to carry out, while motivating them to practice sports for the enjoyment and satisfaction of experiencing positive emotions and feelings, rather than to achieve an award, demonstrate ability, or seek social approval from their peers.

## Figures and Tables

**Figure 1 healthcare-11-00568-f001:**
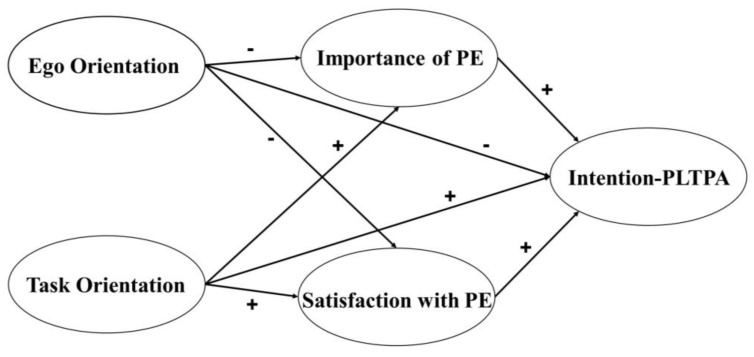
Hypothesized model. Note: dashed lines represent non-significant relationships. Note: PE = Physical Education; Intention-PLTPA = intention to partake in leisure time physical activity.

**Figure 2 healthcare-11-00568-f002:**
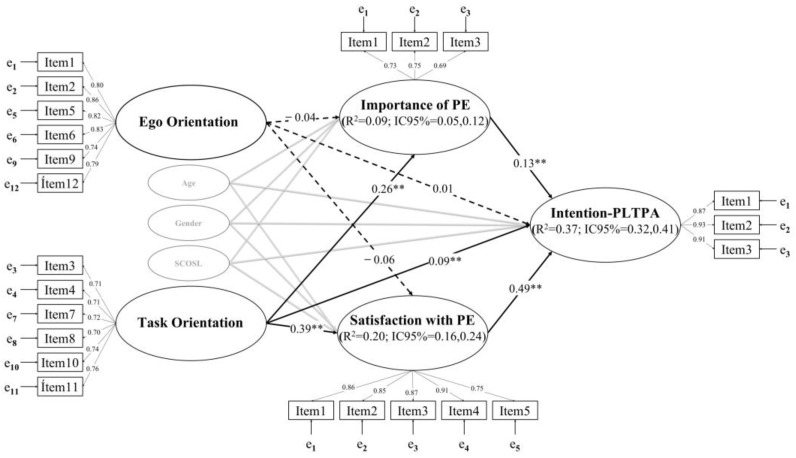
Predictive relationships of dispositional goal orientations on the intention to partake in leisure time physical activity through the mediation of the importance and usefulness of Physical Education and satisfaction/fun with Physical Education. Note: ** *p* < 0.01; SCOSL = sports competition outside of school lessons; PE = Physical Education; Intention-PLTPA = intention to partake in leisure time physical activity; R^2^ = explained variance; CI = confidence interval. Dashed lines represent non-significant relationships. The SEM was controlled for gender, age, and sports competition outside school lessons during the week.

**Table 1 healthcare-11-00568-t001:** Descriptive statistics and correlation between variables.

Variable	Range	*M*	*SD*	Q1	Q2	2	3	4	6
1. Task-orientation	1–5	4.18	0.68	−1.20	1.27	0.34 **	0.18 **	0.32 **	0.28 **
2. Ego-orientation	1–5	3.25	1.13	−0.22	−0.59		0.09 **	0.15 **	0.15 **
3. Importance of PE	1–4	3.05	0.77	−0.79	0.36			0.45 **	0.3 **
4. Satisfaction with PE	1–5	4.19	0.93	−1.08	1.03				0.56 **
5. Intention-PLTPA	1–7	4.97	1.85	−0.64	−0.74				

Note. ** The correlation is significant at the 0.01 level. *M* = mean; *SD* = standard deviation; Q1 = skewness; Q2 = Kurtosis; PE = Physical Education; Intention-PLTPA = intention to partake in leisure time physical activity.

**Table 2 healthcare-11-00568-t002:** Estimation of significant standardized parameters and statistics of the mediation model.

	Independent Variable	Dependent Variable	Mediator	β	SE	95%CI	
						Inf	Sup
Direct effects							
	Task orientation	Importance PE		0.26 **	0.04	0.19	0.33
	Task orientation	Intention-PLTPA		0.09 **	0.03	0.04	0.154
	Task orientation	Satisfaction PE		0.39 **	0.04	0.33	0.45
	Importance PE	Intention-PLTPA		0.13 **	0.04	0.08	0.20
	Satisfaction PE	Intention-PLTPA		0.49 **	0.04	0.42	0.54
Indirect effects							
	Task orientation	Intention-PLTPA	Importance PE	0.05 *	0.05	0.02	0.08
	Task orientation	Intention-PLTPA	Satisfaction PE	0.19 **	0.06	0.12	0.25
Total Effects							
	Task orientation	Intention-PLTPA	Importance PE	0.14 *	0.04	0.05	0.15
	Task orientation	Intention-PLTPA	Satisfaction PE	0.28 *	0.04	0.23	0.36

Note. β = estimation of standardized parameters; SE = standard error; 95%CI = 95% confidence interval; Inf = lower limit of 95%CI; Sup = upper limit of 95%CI; PE = Physical Education; Intention-PLTPA = intention to partake in leisure time physical activity; ** *p* < 0.01; * *p* < 0.05.

## Data Availability

The data presented in this study are available on request from the corresponding author. The data are not publicly available due to privacy.

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
