# Peer review of "Goal Orientations of Secondary School Students and Their Intention to Practise Physical Activity in Their Leisure Time: Mediation of Physical Education Importance and Satisfaction"

_healthcare, 2023, doi:10.3390/healthcare11040568_

Round 1

Reviewer 1 Report

Thank you for the opportunity to review this manuscript. This study found that PE importance and satisfaction play a mediating role in the relationship between task orientation and intention to engage in leisure-time physical activity. Overall, it is an appropriate study with a decent sample size that could be beneficial to students in Spain. However, the manuscript needs to be organized better, particularly for the introduction part. Furthermore, a few more issues should be discussed in the limitation.

Here are some specific comments:

1. Abstract, line 17, “observational” was inappropriate. The study used students’ self-reported questionnaires to obtain data. 

2. Abstract, line 18, please clarify if the term “Physical Education students” was referring to students with a major in Physical Education. If not, please revise.

3. Introduction, line 28, I assume the Organic Law is part of the educational legislation in Spain. Please state it clearly. Also, further information should be given to let readers understand the content of the law and how it relates to PE. As the potential readers will be located worldwide, they may not understand the situation in Spain. Please provide a detailed background for them, and therefore, readers will be able to know the significance of understanding PE in Spain.

4. Lines 53-73, the author partially adopted the Theory of Planned Behavior (TPB) in their study. It is interesting that two of their variables, importance and satisfaction, are somewhat within or related to the construct of attitude in TPB. The authors may consider revising these paragraphs to present a more comprehensive introduction.

5. Lines 74-85, it is a bit confusing here. I understand that the authors tried to bring up one of the variables, satisfaction. However, it seems that the Subjective Well-Being Theory did not play a significant role in this situation. Specifically, as the authors mentioned, it is "a theoretical construct that analyses the perception of human beings regarding their satisfaction", I wonder if you were trying to adopt this theory to measure the satisfaction? Namely, is the instrument SSI-EF development based on this theory? If yes, please provide further information. Otherwise, please modify the introduction and consider other ways to introduce the satisfaction variable.

6. After reading the introduction, I have a major concern about the logical flow, whether the importance of and satisfaction with PE should be directly linked with Intention-PLTPA. For example, importance of PE and satisfaction with PE are more reasonable to associate with the intention to perform physical activity in PE classes. The problem is that you made the assumption that it could simply be translated into leisure-time physical activity without sufficient introduction. Furthermore, it feels like the manuscript did not clearly distinguish between PE and leisure-time physical activity, and somewhat used these two terms interchangeably. Therefore, more discussion should be given about their relationships. I hope the authors could revise the introduction in a more comprehensive and logical way.

7. Apart from age and gender, are there any other demographic variables collected? For example, their past grades in PE; the frequency of PE class (which could be different across grades); if some of them major in PE (in some countries, senior students need to select a major)? We should be aware of these potential confounding factors.

8. For Instruments, please also introduce how to interpret the score for each scale, e.g., a higher score indicates higher orientation, satisfaction, etc.

9. Line 136, “Satisfaction with Physical Education (SSI-EF)”. I think this is not the scale name of SSI-EF, please revise.

10. For 2.3. Procedure, I suggest moving this section after participants and before instruments.

11. Line 200, the authors controlled for gender in the SEM model, however, it should be mentioned earlier in the text, i.e., in the data analysis section.

12. For Figure 2, please mention that it is controlled for gender in the figure notes.

13. Lines 246-247, “the present study show that the total effects of goal orientation on the intention to practice physical activity in one’s leisure time are increased…”. It is more accurate to use “task orientation” instead of “goal orientation”. Please also check if there are similar errors throughout the manuscript.

14. There are more limitations that should be mentioned. First, this study only measured the intention, not the actual behaviors. The concept of "intention-behavior gap" refers to the fact that intentions do not always translate into behaviors. As a result, future studies, preferably longitudinal, should include physical activity in the measurement

For the intention-behavior gap, the following articles may help:
General discussion: 
https://doi.org/10.3389/fpsyg.2022.923464
Examples of the gap in physical activity studies: 
https://doi.org/10.3390/ijerph17010064
https://doi.org/10.1111/bjhp.12032

15. Lines 272-274, “we should note the cross-sectional research design, that the measurements and the results obtained were carried out via questionnaires, and that the sample was not representative.”. Please elaborate on these limitations, i.e., what potential problems they may cause. 

Author Response

We thank the reviewers for his/her constructive comments and his/her thorough revision of the manuscript. Below we answer his/her questions and concerns, including explicitly the changes made in the manuscript as well.

Reviewer 1

Thank you for the opportunity to review this manuscript. This study found that PE importance and satisfaction play a mediating role in the relationship between task orientation and intention to engage in leisure-time physical activity. Overall, it is an appropriate study with a decent sample size that could be beneficial to students in Spain. However, the manuscript needs to be organized better, particularly in the introduction part. Furthermore, a few more issues should be discussed in the limitation.

Here are some specific comments:

Comment-1. Abstract, line 17, “observational” was inappropriate. The study used students’ self-reported questionnaires to obtain data.

  • Response: Suggestions by the reviewer was made.

Comment-2. Abstract, line 18, please clarify if the term “Physical Education students” was referring to students with a major in Physical Education. If not, please revise.

  • Response: Suggestions by the reviewer was made and “Physical Education” was erased. Now, you can read: “In total, 2,102 secondary school students participated…”

Comment-3. Introduction, line 28, I assume the Organic Law is part of the educational legislation in Spain. Please state it clearly. Also, further information should be given to let readers understand the content of the law and how it relates to PE. As the potential readers will be located worldwide, they may not understand the situation in Spain. Please provide a detailed background for them, and therefore, readers will be able to know the significance of understanding PE in Spain.

  • Response: (see lines 28-42) The reviewer's suggestions have been addressed: “Physical Education (PE) in Spain has significant importance in Organic Law 3/2020, of December 29 [1]. This law regulates non-university educational teaching in the different age groups and encourages the education system and the PE to help people incorporate sports practice into their lives and thus contribute to an active and healthy lifestyle. In fact, this law adds an additional provision that was not contemplated in the previous one, Organic Law 2/2006, of May 3 [2]. This provision refers to the promotion of physical activity and healthy eating so that they are part of the behavior of children and young people. This line mentions that the Administrations will promote the daily practice of sports and physical exercise by students during the school day, under the terms and conditions that, following the recommendations of the government agencies, guarantee adequate development to promote a healthy and autonomous life, to promote healthy eating habits and active mobility, reducing sedentary lifestyle [1]. Despite this, adolescence is a stage in which the practice of physical activity is still largely neglected; in addition, there is a high rate of physical inactivity in this population [3,4], which is because it is an important issue to address in the scientific field.”

Comment-4. In lines 53-73, the author partially adopted the Theory of Planned Behavior (TPB) in their study. It is interesting that two of their variables, importance, and satisfaction, are somewhat within or related to the construct of attitude in TPB. The authors may consider revising these paragraphs to present a more comprehensive introduction.

  • Response: (see Line 71-80): “However, it should be noted that there is a gap between intention and behavior, that is, not every intention to perform a behavior becomes a behavior itself [17]. Regarding the intention to be physically active, studies have shown this gap exists between the intention to be physically active and developing a positive behavior towards physical activity [17,18]. Although the intention is the proximal antecedent of the enactment of behavior [19]. In this line, the intention of adolescents to practice physical activity outside school hours has been related both to the importance and usefulness they give to PE [20] and to the satisfaction they experience with the subject [10]. For this reason, we consider important to investigate how we can act from the PE subject, in order to increase the intention of adolescents to be physically active in their free time.”

Comment-5. Lines 74-85, it is a bit confusing here. I understand that the authors tried to bring up one of the variables, satisfaction. However, it seems that the Subjective Well-Being Theory did not play a significant role in this situation. Specifically, as the authors mentioned, it is "a theoretical construct that analyses the perception of human beings regarding their satisfaction", I wonder if you were trying to adopt this theory to measure satisfaction? Namely, is the instrument SSI-EF development based on this theory? If yes, please provide further information. Otherwise, please modify the introduction and consider other ways to introduce the satisfaction variable.

  • Response: The Theory of Subjective Well-being by Diener and Emmons (1985) is a theoretical construct that analyzes the perception of human beings regarding their satisfaction with life in general, as well as in specific areas of life such as family, friends, school, sports practice, etc., specific area for adolescents in which they can make judgments in terms of satisfaction/fun or dissatisfaction/boredom based on how they learn and experience the subject (Baños, 2020). See lines 96-100: “In this vein, Balaguer et al. [25] designed an instrument that measured satisfaction with the school, which was later adapted to the PE subject by Baena-Extremera et al. [26], in order to measure the satisfaction and enjoyment experienced by students with PE, in this specific area of adolescent life. This enjoyment with PE acts as an excellent predictor of both the intention to practice physical activity outside PE lessons [9] (…)”

Comment-6. After reading the introduction, I have a major concern about the logical flow, whether the importance of and satisfaction with PE should be directly linked with Intention-PLTPA. For example, the importance of PE and satisfaction with PE are more reasonable to associate with the intention to perform physical activity in PE classes. The problem is that you made the assumption that it could simply be translated into leisure-time physical activity without sufficient introduction. Furthermore, it feels like the manuscript did not clearly distinguish between PE and leisure-time physical activity, and somewhat used these two terms interchangeably. Therefore, more discussion should be given about their relationships. I hope the authors could revise the introduction in a more comprehensive and logical way.

  • Response: Modifications have been made to the introduction to improve the wording. We have also tried to make it clear that the intention variable in our study refers to the intention to be active outside of school hours. In addition, the SEM has been controlled with the variable of whether or not students perform physical activity (competitions) outside school hours.

Comment-7. Apart from age and gender, are there any other demographic variables collected? For example, their past grades in PE; the frequency of PE class (which could be different across grades); if some of them major in PE (in some countries, senior students need to select a major)? We should be aware of these potential confounding factors.

  • Response: Thank you for this comment. Initial socio-demographic values have been added: percentage of students per course, ad rate of dropout, for example. Likewise, it was clarified that PE sessions are compulsory for all students, and they attend two sessions of 60 minutes / per week. In addition, 28.3% of the students participate in sporting competitions outside of school during the week (Section 2.1. Design and Participants) In addition, to avoid possible differences regarding gender, age, and physical activity outside of school, the SEM has been controlled by gender, age, and sporting competition outside of school during the week. This way, the results haven't been conditioned by these variables.

Comment-8. For Instruments, please also introduce how to interpret the score for each scale, e.g., a higher score indicates higher orientation, satisfaction, etc.

  • Response: Suggestions by the reviewer was made.

Comment-9. Line 136, “Satisfaction with Physical Education (SSI-EF)”. I think this is not the scale name of SSI-EF, please revise.

  • Response: Suggestions by the reviewer was made. The scale name is SSI-PE.

Comment-10. For 2.3. Procedure, I suggest moving this section after participants and before instruments.

  • Response: Suggestions by the reviewer was made.

Comment-11. Line 200, the authors controlled for gender in the SEM model, however, it should be mentioned earlier in the text, i.e., in the data analysis section.

  • Response: As the reviewer suggests, this information has been added in the data analysis section (lines 212-213).

Comment-12. For Figure 2, please mention that it is controlled for gender in the figure notes.

  • Response: As the reviewer suggests, this information has been added in the figure-2 notes (see lines 274-275).

Comment-13. Lines 246-247, “the present study show that the total effects of goal orientation on the intention to practice physical activity in one’s leisure time are increased…”. It is more accurate to use “task orientation” instead of “goal orientation”. Please also check if there are similar errors throughout the manuscript.

  • Response: We use “goal orientation” to refer to "dispositional goal orientation" (i.e., task orientation and ego orientation). Even the scale consists of 12 items that measure the students' dispositional goal orientations in PE classes: task orientation (six items) and ego orientation (six items). We have clarified throughout the manuscript and added "dispositional goal orientation" instead of "goal orientation".

Comment-14. There are more limitations that should be mentioned. First, this study only measured the intention, not the actual behaviors. The concept of "intention-behavior gap" refers to the fact that intentions do not always translate into behaviors. As a result, future studies, preferably longitudinal, should include physical activity in the measurement. 

For the intention-behavior gap, the following articles may help:

General discussion: 

https://doi.org/10.3389/fpsyg.2022.923464
Examples of the gap in physical activity studies:

https://doi.org/10.3390/ijerph17010064
https://doi.org/10.1111/bjhp.12032

  • Response:

Thank you for the recommendation to read the articles, it has been quite useful to me

In lines 328-330, the limitations suggested by the reviewer have been added:

Regarding the limitations, point out that the present study only measured the intention, not the actual behaviors, that is, the intentions of adolescents to be physically active do not always translate into active behaviors.

Lines 338-341, future lines of research have been added:

As future research, we suggest that longitudinal or experimental studies should be carried out, in which it is analyzed over time if the intention to be physically active becomes behaviors, and analyze which are the most influential variables in this transformation.

Comment-15. Lines 272-274, “we should note the cross-sectional research design, that the measurements and the results obtained were carried out via questionnaires, and that the sample was not representative.”. Please elaborate on these limitations, i.e., what potential problems they may cause. 

  • Response:

Lines 338-343, the reviewer's comments have been addressed:

It is worth noting the cross-sectional research design since when it is carried out at a determined time, it is not possible to analyze the behavior of the subjects during a determined time. Another limitation of the study was that the measurements and the results obtained were carried out through questionnaires, this could condition the responses of the participants due to the search for social desirability, causing a bias in the research. Finally, it is worth mentioning that the sample was not representative, so the results cannot be generalized to Spanish adolescents.

Reviewer 2 Report

I have reviewed the manuscript titled "Secondary students' goal orientations and their intention to practice physical activity in their free time: mediating the importance and satisfaction of Physical Education" for its consideration for publication in Healthcare journal.

It is a very interesting study that will probably encourage other authors to carry out more studies. Congratulations to the authors. However, prior to its potential publication, there are some issues that in my opinion should be clarified.

It would be interesting if the authors shared initial values, which they surely must have, since data would be collected from the 2102 schoolchildren: Why have they ruled out putting them? Didn't you find it interesting?

The authors talk about the characteristics of the schoolchildren on page 3 lines 124-125, is it possible that there were differences between them regarding levels of physical activity by age, or that differences were found between male or female schoolchildren?

I believe that these clarifications may be essential so that the result is not conditioned, since the intention to practice a sport may be out, if it is already a regular daily activity.

Author Response

We thank the reviewers for his/her constructive comments and his/her thorough revision of the manuscript. Below we answer his/her questions and concerns, including explicitly the changes made in the manuscript as well.

It is a very interesting study that will probably encourage other authors to carry out more studies. Congratulations to the authors. However, prior to its potential publication, there are some issues that in my opinion should be clarified.

Comment-1. It would be interesting if the authors shared initial values, which they surely must have, since data would be collected from the 2102 schoolchildren: Why have they ruled out putting them? Didn't you find it interesting? The authors talk about the characteristics of the schoolchildren on page 3 lines 124-125, is it possible that there were differences between them regarding levels of physical activity by age, or that differences were found between male or female schoolchildren? I believe that these clarifications may be essential so that the result is not conditioned, since the intention to practice a sport may be out if it is already a regular daily activity.

  • Response: Thank you for this comment. Initial socio-demographic values have been added: percentage of students per course, ad rate of dropout, for example. Likewise, it was clarified that PE sessions are compulsory for all students, and they attend two sessions of 60 minutes / per week. In addition, 28.3% of the students participate in sporting competitions outside of school during the week (Section 2.1. Design and Participants) In addition, to avoid possible differences regarding gender, age, and physical activity outside of school, the SEM has been controlled by gender, age, and sporting competition outside of school during the week. This way, the results haven't been conditioned by these variables.

Reviewer 3 Report

- Interesting study on SEM.

- The use of SEM is justified given the authors are trying to measure concepts like task-orientation and ego orientation which are difficult to quantify otherwise.

- Would it be possible to provide the questionnaires for the surveys especially in English? Even Spanish would suffice.

- Tackles an important issue of adolescent inactivity and obesity that is rampant in all developed countries with the advent of electronic devices.

Author Response

We thank the reviewers for his/her constructive comments and his/her thorough revision of the manuscript. Below we answer his/her questions and concerns, including explicitly the changes made in the manuscript as well.

Comment-1. Interesting study on SEM.

  • Response: Thank you very much.

Comment-2. The use of SEM is justified given the authors are trying to measure concepts like task-orientation and ego orientation which are difficult to quantify otherwise.

  • Response: Thank you very much.

Comment-3.- Would it be possible to provide the questionnaires for the surveys especially in English? Even Spanish would suffice.

  • Response: You can find the scales for the surveys in the following references. Any reader could easily find the scales.

POSQ: Granero-Gallegos, A.; Baena-Extremera, A.; Gómez-López, M.; Abraldes. A. Psychometric Study and the Prediction of the Importance of Physical Education from Goal Guidance (“Perception of Success Questionnaire – POSQ”). Psicol.-Reflex. Crit., 2014, 27(3), 443-451. https://doi.org/10.1590/1678-7153.201427304

 IPE: Moreno, J. A.; González-Cutre, D.; Ruiz, L.M. (2009). Self-determined motivation and physical education importance. Human Mov., 2009, 10(1), 5-11. https://doi.org/10.2478/v10038-008-0022-7

 SSI_PE: Baena-Extremera, A.; Granero-Gallegos, A.; Bracho-Amador, C.; Pérez-Quero, F.J. Spanish Version of the Sport Satisfaction Instrument (SSI) Adapted to Physical Education. Rev. Psicodidact., 2012, 17(2), 377-396. https://10.1387/RevPsicodidact.4037

 Intention-PLTPA: Granero-Gallegos, A.; Baena-Extremera, A.; Pérez-Quero, F.J.; Ortiz-Camacho, M.M.; Bracho-Amador, C. Spanish validation of the scale «intention to leisure-time in partake physical activity». Retos2014, 26, 40–45. https://doi.org/10.47197/retos.v0i26.34392

Comment-4.- Tackles an important issue of adolescent inactivity and obesity that is rampant in all developed countries with the advent of electronic devices.

  • Response: Thank you very much.

Reviewer 4 Report

In the manuscript „Goal orientations of secondary school students and their intention to practise physical activity in their leisure time: mediation of Physical Education importance and satisfaction” the Authors tried to assess the mediating role of satisfaction with physical education and the perceived importance and usefulness of physical education between the goal orientations of secondary school students and their intention to practise physical activity in their leisure time. This is a cross-sectional study, which included 2102 secondary school students from Andalusia. The research topic addressed in this study is of great interest and important.

Generally, the manuscript provides valuable information. However, I have some remarks/questions.

I recommend adding information about reliability of tools in the Instruments section.

There is a lack of description of questions on sociodemographic characteristics. Did you ask about them, right?

How long did it take to complete the questionnaire?

Was the teacher present during the survey, or did she/he leave? 

Author Response

We thank the reviewers for his/her constructive comments and his/her thorough revision of the manuscript. Below we answer his/her questions and concerns, including explicitly the changes made in the manuscript as well.

In the manuscript „Goal orientations of secondary school students and their intention to practise physical activity in their leisure time: mediation of Physical Education importance and satisfaction” the Authors tried to assess the mediating role of satisfaction with physical education and the perceived importance and usefulness of physical education between the goal orientations of secondary school students and their intention to practise physical activity in their leisure time. This is a cross-sectional study, which included 2102 secondary school students from Andalusia. The research topic addressed in this study is of great interest and important.

Generally, the manuscript provides valuable information. However, I have some remarks/questions.

Comment-1.- I recommend adding information about reliability the tools in the Instruments section.

  • Response: Suggestions by the reviewer was made. The reliability of tools has been added in the instruments section and removed in Table 1. As well, the CFA of each tool has been added in the instrument section.

Comment-2.- There is a lack of description of questions on sociodemographic characteristics. Did you ask about them, right?

  • Response: Thank you for this comment. Initial socio-demographic values have been added: percentage of students per course, ad rate of dropout, for example. Likewise, it was clarified that PE sessions are compulsory for all students, and they attend two sessions of 60 minutes / per week. In addition, 28.3% of the students participate in sporting competitions outside of school during the week (Section 2.1. Design and Participants) In addition, to avoid possible differences regarding gender, age, and physical activity outside of school, the SEM has been controlled by gender, age, and sporting competition outside of school during the week. This way, the results haven't been conditioned by these variables.

Comment-3.- How long did it take to complete the questionnaire?

  • Response: The following information has been added in the section “procedure”: “The students had 15 minutes to answer the questionnaire and the PE teacher was not present in the classroom”.

Comment-4.- Was the teacher present during the survey, or did she/he leave? 

  • Response: The following information has been added in the section “procedure”: “The students had 15 minutes to answer the questionnaire and the PE teacher was not present in the classroom”.

Round 2

Reviewer 1 Report

Thanks for your detailed reply to my comments on this work.

The manuscript has been greatly improved and all my concerns have been addressed.

I have no further comments.

Reviewer 2 Report

I have revisited the manuscript for consideration for publication in Healthcare journal. Congratulations to the authors, they have resolved the issues that in my opinion needed to be clarified.